# Genome-Wide Identification of *BrCMF* Genes in *Brassica rapa* and Their Expression Analysis under Abiotic Stresses

**DOI:** 10.3390/plants13081118

**Published:** 2024-04-17

**Authors:** Luhan Chen, Xiaoyu Wu, Meiqi Zhang, Lin Yang, Zhaojing Ji, Rui Chen, Yunyun Cao, Jiabao Huang, Qiaohong Duan

**Affiliations:** 1College of Horticulture Science and Engineering, Shandong Agricultural University, Tai’an 271000, China; clh2447674277@hotmail.com (L.C.); 2020110271@sdau.edu.cn (X.W.); zmqamm@163.com (M.Z.); yanglin@sdau.edu.cn (L.Y.); 2022120394@sdau.edu.cn (Z.J.); chendian0605@163.com (R.C.); caoyunyun@sdau.edu.cn (Y.C.); 2Institute of Vegetable Science, Zhejiang University, Hangzhou 310058, China

**Keywords:** *CCT* gene family, *CMF*, abiotic stress, *Brassica rapa*, genome-wide identification

## Abstract

*CCT MOTIF FAMILY* (*CMF*) genes belong to the *CCT* gene family and have been shown to play a role in diverse processes, such as flowering time and yield regulation, as well as responses to abiotic stresses. *CMF* genes have not yet been identified in *Brassica rapa*. A total of 25 *BrCMF* genes were identified in this study, and these genes were distributed across eight chromosomes. Collinearity analysis revealed that *B. rapa* and *Arabidopsis thaliana* share many homologous genes, suggesting that these genes have similar functions. According to sequencing analysis of promoters, several elements are involved in regulating the expression of genes that mediate responses to abiotic stresses. Analysis of the tissue-specific expression of *BrCMF14* revealed that it is highly expressed in several organs. The expression of *BrCMF22* was significantly downregulated under salt stress, while the expression of *BrCMF5*, *BrCMF7*, and *BrCMF21* was also significantly reduced under cold stress. The expression of *BrCMF14* and *BrCMF5* was significantly increased under drought stress, and the expression of *BrCMF7* was upregulated. Furthermore, protein–protein interaction network analysis revealed that *A. thaliana* homologs of *BrCMF* interacted with genes involved in the abiotic stress response. In conclusion, *BrCMF5*, *BrCMF7*, *BrCMF14*, *BrCMF21*, and *BrCMF22* appear to play a role in responses to abiotic stresses. The results of this study will aid future investigations of *CCT* genes in *B. rapa*.

## 1. Introduction

*CMF* genes comprise a subfamily of the *CCT* gene family and have diverse functions. The *CCT* gene family contains conserved CCT motifs (CONSTANS, CO-like, and TOC1), and they were first described from the *A. thaliana CONSTANS* gene [1]; with these genes having been shown to be involved in flowering regulation and the response to photoperiod [2]. Several studies have indicated that the *CCT* gene family can be divided into four subfamilies based on conserved domains: (1) the *CONSTANS-like* (*COL*) subfamily, which contains one or two zinc finger B-boxes (BBOX) and a CCT domain; (2) the *Pseudo-response regulator* (*PRR*) subfamily, which contains the CCT domain and a response regulator domain (REC) [3]; (3) *CCT MOTIF FAMILY* (*CMF*) subfamily, which contains only one conserved CCT domain [4]; and (4) the *ZINC-FINGER PROTEIN EXPRESSED IN INFLORESCENCE MERISTEM (ZIM)* subfamily, which is referred to as *CMF* in some studies, which contains the Tify, CCT, and ZnF_GATA domains [5,6,7]. COL proteins are primarily involved in the regulation of flowering time and the biological clock. PRR proteins are primarily involved in circadian rhythm regulation, and ZIM proteins are involved in the elongation of the petiole and hypocotyl cells [6]. The activities of the proteins encoded by *CMF* genes are diverse, but more work is needed to clarify the functions of these genes. According to phylogenetic analysis of *CMF* genes in Gramineae, the BBOX domain of *COL* has been lost over evolutionary time in the lineage leading to *CMF* genes, which only contain one CCT domain [4].

*CMF* genes have been identified in various plants, including *Arabidopsis thaliana* [4], *Triticum aestivum* L. [8], *Oryza sativa* [7], *Oryza rufipogon* [9], *Medicago truncatula* [10], *Setaria italica* [11], *Glycine max* [12], *Zea mays* [13], *Aegilops tauschii* [14], *Cajanus cajan L.* [15], and *Populus trichocarpa* [16]. Several *CMF* genes in gramineous plants have been shown to play a role in circadian rhythm regulation, as well as heading and flowering time regulation. These genes also play a role in responses to abiotic stress in *G. max*, *M. truncatula*, *S. italica*, and *A. tauschii*. For example, the expression of *MtCCT*s has been shown to be altered under salt and drought stress following treatment with abscisic acid (ABA); thus, *MtCCT*s have been hypothesized to play a role in responses to abiotic stress through an ABA-dependent or an ABA-independent mechanism [10]. The expression of *GmCMF06* and *GmCM07* in *G. max* is upregulated following exposure to drought stress [12]. During drought stress, the expression of most *CMF* genes in *S. italica* was upregulated 40-fold. At 12 h of drought exposure, the expression of *SiCCT3* was significantly upregulated, and *SiCCT31* expression was significantly altered by low temperature and salt stress. *SiCCT31* expression was significantly upregulated more than 8-fold under low-temperature stress, but its expression was significantly downregulated under salt stress [11]. In *A. tauschii*, the expression of *AetCCT16* and *AetCCT20*, which belong to the *CMF* family, was downregulated following drought and NaCl treatment [14]. These studies suggest that *CMF* genes play a role in responses to abiotic stress. The roles of different types of CMFs in regulating chloroplast development, regulating genes expression, and playing a role in stress-induced have been identified in *A. thaliana*. For example, CIA2 (*AtCMF14*) and *CIL* (*AtCMF9*) regulate chloroplast development of *Arabidopsis thaliana* [17], ASML2 (*AtCMF8*) regulates the expression of glycogen-inducible genes [18], FITNESS (*AtCMF3*) plays a key role in regulating reactive oxygen levels and defense responses [19], and GHD7 (*OsCMF8*) regulates tassel date and stress tolerance in *O. sativa* [20]. *CMF* genes thus play a role in regulating diverse life activities.

Abiotic stress has a major effect on the yield of *Brassica rapa*, which is an economically important crop, and climate change is likely to increase the frequency and intensity of weather extremes [21]. *B. rapa* has traditionally been grown in arid and semi-arid regions of northern Asia, where droughts, salinity, and cold stress are common. These adverse environmental conditions have led to declines in the yield of *B. rapa* [22]. The growth of the human population has increased the demand for vegetables, and various types of abiotic stress negatively affect *B. rapa* yields. There is thus a need to identify genes involved in responses to abiotic stress to enhance the stress resistance and yields of *B. rapa*.

*CMF* genes regulate various biological activities, but the *BrCMF* genes of *B. rapa* have not been identified to date. In this study, we identified 25 *BrCMF* genes at the genome-wide level. We then analyzed the phylogenetic relationships, collinearity, and transcriptome expression profiles of these genes, as well as the physicochemical properties of their encoded proteins and protein interaction relationships. We explored the relationship between this family of genes and abiotic stress. The purpose of this study was to provide new insights that could be used to aid future investigations of *CMF* genes and efforts to breed *B. rapa* varieties with enhanced stress resistance.

## 2. Results

### 2.1. Identification and Analysis of BrCMFs

A total of 26 BrCMF genes were identified based on comparison of homologous genes of the known AtCMF family with those in the *B. rapa* genome database. A gene member without the CCT domain was identified and omitted, which yielded a total of 25 predicted CMF genes, exceeding the number of CMF genes reported in Oryza sativa (19) [7], Oryza rufipogon (19) [9], Zea mays (14) [13], Setaria italica (19) [11], Aegilops tauschii (12) [14], Medicago truncatula (18) [10], Glycine max (19) [12], *Cajanus cajan* L. (12) [15], and Populus trichocarpa (14) [16]. These genes were named according to their chromosomal distribution. ATCMF IDs, homologous gene names, chromosome positions, pI, MW, protein length (aa), subcellular localization, and their names (and alternate names) are listed in Table 1. The pI ranged from 4.45 (BrCMF11) to 9.67 (BrCMF24), the MW ranged from 17,412.59 Da (BrCMF2) to 60,150.03 Da (BrCMF24), and the protein length ranged from 153 aa (BrCMF2) to 535 aa (BrCMF24). The protein length was linearly positively correlated with the MW. All the predicted proteins were predicted to be located in the nucleus, among which a few were also predicted to be located in the cytosol, mitochondria, and chloroplast. CMF genes, which are mostly known for their role in regulating flowering time, might play key roles in the nucleus.

### 2.2. Chromosomal Localization, Synteny, and Phylogenetic Analysis of BrCMFs

*BrCMF* genes were unevenly distributed across eight chromosomes of *B. rapa*. *BrCMF25* was the only gene that was not mapped to one of the 10 chromosomes, and it was located on Scaffold000164, an unknown chromosome, and showed no collinearity with any other genes. The greatest number of *BrCMF* genes was located on Chromosome 10 (6), and the lowest number of *BrCMF* genes were located on Chromosome 1 (1). Chromosomes 2, 3, 4, 5, 8, and 9 contained four, four, two, two, three, and two *BrCMF* genes, respectively; no *BrCMF* genes were detected on Chromosomes 6 and 7 (Figure 1A).

To study the collinearity relationships among *BrCMF* genes, we examined the homology of *CMF* genes between *B. rapa* and *A. thaliana* and between *B. rapa* and *O. sativa* (Figure 1B). Collinearity was highest for *CMF* genes between *B. rapa* and *A. thaliana*. We divided *CMF* genes from *B. rapa*, *A. thaliana*, and *O. sativa* into Groups 1–4 based on their genetic relationships. G2 (Group 2) contained the most predicted *BrCMF* genes (*BrCMF1*, *BrCMF2*, *BrCMF4*, *BrCMF6*, *BrCMF9*, *BrCMF15*, *BrCMF18*, *BrCMF19*, *BrCMF22*, and *BrCMF24*), followed by G3 (*BrCMF8*, *BrCMF11*, *BrCMF12*, *BrCMF13*, *BrCMF16*, *BrCMF20*, and *BrCMF25*), G1 (*BrCMF5*, *BrCMF7*, *BrCMF14*, and *BrCMF21*), and G4 (*BrCMF3*, *BrCMF10*, *BrCMF17*, and *BrCMF23*) (Figure 1C). *CMF* genes in *B. rapa* were more closely related to *A. thaliana* than to *O. sativa*.

The similarity in the expression patterns of genes in organs was high among highly homologous genes. For example, the phylogenetic tree indicated that *BrCMF15* and *BrCMF19* were closely related to *AtCMF3* (FITNESS), suggesting that these two genes might have functionally diversified in *B.rapa*. In addition, the expression patterns of these two genes were similar in various organs. *BrCMF2*, *BrCMF6*, and *BrCMF24*, which are homologous to *AtCMF11*, were expressed at low levels. Significant differences in the expression of *BrCMF22*, *BrCMF1*, and *BrCMF9* were observed, presumably because *AtCMF9* (*CIL*) was more closely related to *BrCMF9* and *BrCMF1* than to *BrCMF22*. This explains the similarity in the expression patterns of *BrCMF9* and *BrCMF1*, as well as the contrasting expression patterns of *BrCMF22* in different organs.

### 2.3. Motif Analysis, Gene Structure, and Conserved Domain Analysis of BrCMFs

Using TBtools [23], we were able to obtain a conserved domain map of *BrCMF* genes, and the conserved domain diagram of the 25 predicted *BrCMF* genes is shown in Figure 2B. Each *BrCMF* has a conserved CCT domain. *BrCMF18*, *BrCMF24*, and *BrCMF15* possess other conserved motifs in addition to the CCT domain. *BrCMF18* contains a BBOX domain, which indicates that this gene is a member of the *COL* subfamily. Both the *COL* and *CMF* subfamilies contain CCT domains, and the *COL* family contains one to two BBOX domains, while *CMF* family members do not contain BBOX domains. Therefore, we predict that *BrCMF18*’s domain has been evolutionarily acquired, given that the BBOX domain in *COL* genes might have been lost, and this led to the origin of the *CMF* family [4,9].

The motif diagram of *BrCMF* genes revealed that all genes have Motif 1 and Motif 2, and *BrCMF22* has the most motifs (6). *BrCMF1*, *BrCMF4*, *BrCMF14*, *BrCMF5*, *BrCMF7*, *BrCMF21*, *BrCMF20*, *BrCMF16*, *BrCMF12*, and *BrCMF11* each have five motifs; *BrCMF18*, *BrCMF15*, *BrCMF23*, *BrCMF3*, *BrCMF10*, and *BrCMF17* only have one or two motifs. There was a close relationship between motif distributions and gene homology. Specifically, highly homologous genes tended to have similar motif distributions, indicating that similar genes have similar functions.

To further clarify the function of genes, we characterized the structure of *BrCMF* genes (Figure 2C). The coding sequence (CDS) represents the functional region of the gene containing exons, and lines indicate the functional region containing introns. The structure of 25 predicted genes varies greatly: *BrCMF15* and *BrCMF24* contain the most introns, while *BrCMF18*, *BrCMF13*, and *BrCMF25* do not contain introns. Further evidence of the functional differentiation and specificity of the gene families is revealed by these structural differences.

### 2.4. Cis-Acting Element Analysis of BrCMFs

We identified 24 *cis*-acting elements in a 2000 bp 5′ region of *BrCMF* genes; *cis*-acting elements in the promoter of the *BrCMF25* gene were not characterized. The functions of *cis*-acting elements can be divided into three categories: growth and development, phytohormone response, and stress response. The number and distribution of various *cis*-acting elements for each gene are shown in Figure 3.

In the first category (growth and development), all genes contain light-responsiveness elements, which is consistent with the role of *CCT* genes in regulating photoperiodic flowering. Many circadian rhythm-related elements were detected in *BrCMF1*, *BrCMF3*, *BrCMF7*, *BrCMF11*, and *BrCMF14*. In addition, some elements involved in the seed-specific regulation of endosperm expression and zein metabolism regulation were observed.

In the phytohormone response group, genes contained five different types of elements: ABA-responsiveness, auxin-responsiveness, gibberellin-responsiveness, MeJA-responsiveness, and salicylic acid-responsiveness elements. Most of these genes contained elements that are MeJA-responsive, and *BrCMF8* contained the most *cis*-acting elements (10). ABA and gibberellin-responsive elements were present in 16 and 14 *BrCMF* genes, respectively. Most *BrCMF* genes contained three–four *cis*-acting elements, and *BrCMF4* did not contain phytohormone-responsive *cis*-acting elements.

The promoters of 10, 11, 21, and 16 *BrCMF* genes had defense and stress-responsiveness, drought-inducibility, anaerobic induction, and low-temperature-responsiveness elements, respectively. Most genes contain two–three types cis-acting elements in the Stress response group, suggesting that some of these genes play an important role in stress responses.

### 2.5. Tissue Specific Expression of BrCMFs

To identify the potential roles of *BrCMF*s in abiotic stress responses, we analyzed tissue-specific expression data for callus, flower, leaf, root, silique, and stem tissues of *B. rapa* from the BRAD (http://brassicadb.cn/, accessed on 22 November 2022) and made expression heat maps (Figure 4, Appendix A) to enhance our understanding of their potential functions. In most tissues, the expression of the genes was low; however, the expression of eight genes (*BrCMF5*, *BrCMF7*, *BrCMF10*, *BrCMF14*, *BrCMF15*, *BrCMF19*, *BrCMF21*, and *BrCMF22*) was high in some tissues and low in others. Genes with similar sequences tended to have similar high expression levels. *BrCMF14* and *BrCMF22* are highly expressed in multiple tissues, while the expression of *BrCMF14* was high in callus, flower, root, silique, and stem tissues, and the expression of *BrCMF22* was high in flower, leaf, and stem tissues, suggesting that these two genes play roles in several life activities. Both *BrCMF19* and *BrCMF15* were highly expressed in leaves, and *BrCMF10* was highly expressed in roots, suggesting that they play key roles in diverse life activities. The homologous genes *BrCMF5*, *BrCMF7*, and *BrCMF21* were highly expressed in silique and stem tissue, and *BrCMF7* was highly expressed in flower and leaf tissue.

### 2.6. Transcriptome Analysis of BrCMFs

We compared the expression levels of 24 *BrCMF* genes under cold, drought, and salt stress using transcriptome data (Figure 5A–C, Appendix A). The expression of most genes was low under salt, drought, and cold stress. For example, the expression levels of *BrCMF8*, *BrCMF12*, *BrCMF13*, *BrCMF18*, and *BrCMF19* were always 0 under three stress conditions, corresponding to no height of the columns; however, significant changes were observed in the expression of some genes. Salt treatment decreased the expression of *BrCMF5*, *BrCMF21*, and *BrCMF22* and increased the expression of *BrCMF7* and *BrCMF15*. Following exposure to drought stress, the expression of several genes was significantly downregulated and the expression of *BrCMF7* was upregulated, which suggests that *BrCMF7* plays a special role in the response to drought stress. During cold treatment, the expression of *BrCMF5*, *BrCMF7*, and *BrCMF21* was downregulated; however, the expression of *BrCMF15* and *BrCMF22* was upregulated. Changes in the expression of *BrCMF5*, *BrCMF7*, *BrCMF15*, *BrCMF21*, and *BrCMF22* strongly suggest that they play an important role in the response to abiotic stress.

To confirm our hypothesis, we conducted RT-qPCR analyses of key genes that were inferred to have tissue-specific expression patterns according to RNA-seq analysis, and the screened genes showed a significant downregulation trend within 12 h under both salt and cold stress. The expression levels of *BrCMF22*, *BrCMF5*, *BrCMF7*, and *BrCMF21* inferred by RNA-seq and RT-qPCR analyses were consistent, and they were hypothesized to be involved in salt and cold stress. *BrCMF15* showed a significant upregulation trend in both the transcriptome and RT-qPCR after 7 days of cold treatment (Figure 5C,E, Appendix A), and it was hypothesized that *BrCMF15* might function under prolonged cold stress. Expression patterns (transcriptome and RT-qPCR) were consistent for the majority of genes, with differences in a few gene changes possibly being due to experimental or sampling differences, as has occurred in previous studies [24]. Under drought stress, the expression levels of *BrCMF14* and *BrCMF5* were significantly downregulated more than 3-fold and 10-fold, respectively, and the expression of *BrCMF7* was significantly upregulated more than 3-fold (Figure 5D, Appendix A). These results were consistent with the transcriptome data (Figure 5B, Appendix A), confirming that these genes might be involved in drought-stress-related functions. The expression level of *BrCMF15* significantly increased after 4 h of drought treatment, and, as the treatment time increased, the expression level of *BrCMF15* was sharply downregulated. After 6 h of drought stress, the expression level in RT-qPCR experiments was significantly downregulated (Figure 5D, Appendix A), which is consistent with the transcriptome data (Figure 5B, Appendix A). It proves the correctness of the experimental results. The relative expression of *BrCMFs* was significantly downregulated under short duration cold stress, and the expression of *BrCMF15* of these genes was lowest under such conditions (Figure 5E, Appendix A). Studies have shown that MeJA could promote reactive oxygen species (ROS) scavenging and contribute to the positive feedback regulation of melatonin (MT) to enhance the cold tolerance of plants [25,26], but *BrCMF15* lacks MeJA-response elements. This is consistent with the high expression of the other genes according to the qPCR analysis.

### 2.7. Protein–Protein Interaction (PPI) Network Analysis of BrCMFs

The STRING tool was used to predict interactions between proteins and *A. thaliana* homologous *BrCMF* genes. Four *A. thaliana* homologous genes interacted with multiple abiotic stress proteins, including *AtCMF3* (homologs: *BrCMF15* and *BrCMF19*, Figure 6A), *AtCMF9* (homologs: *BrCMF22*, *BrCMF1*, and *BrCMF9*, Figure 6B), *AtCMF10* (homolog: *BrCMF14*, Figure 6C), and *AtCMF13* (homologs: *BrCMF5*, *BrCMF7*, and *BrCMF21*, Figure 6D).

*At1G07050* (*AtCMF3*) interacted with AT5G23240 and AT5G42900 (COR27)-related proteins; AT5G23240 plays a role in root development by mediating protein folding and preventing the aggregation of proteins in chloroplasts during salt stress. The *COR27* gene plays a role in the response to cold stress by mediating the effects of cold signals on the biological clock, enhancing freezing tolerance, and entraining circadian rhythms [27]. *CIL (AtCMF9)* interacts with *EMB1138*, which encodes plastid-specific enzymes involved in ABA biosynthesis; this plays an important role in maintaining ABA levels and mediating salt stress responses. *AT4G27900* interacted with *AT5G53420* and was associated with PRR7 and ARP6. ARP6 is responsible for mediating ambient temperature responses. Moreover, PRR7 is encoded by a *CCT* gene and has been shown to modulate drought and ABA responses as well as the expression of genes under cold stress [28]. The target gene of PRR7 is regulated by ABA and contains ABA-responsive elements in its upstream region [29], while PRR7 is involved in plant growth and development, regulation of the biological clock, and the photoperiodic flowering response.

## 3. Discussion

*CMF*, *COL*, and *PRR* genes are subfamilies of the *CCT* gene family, and their roles in flowering time regulation have been studied. The *PRR* subfamily and the *COL* subfamily of the *B. rapa CCT* gene family have been identified [30,31], but whole-genome identification and analysis of *BrCMF* genes have not yet been carried out in *B. rapa*. We identified 25 *BrCMF* genes on eight chromosomes in *B. rapa*, which is more than the number of *CMF* genes identified in previously studied plants of this family. Previous studies have demonstrated that *Brassicaceae* underwent three whole genome replication events (WGT) approximately 9–15 million years ago, which greatly increased the number of genomes of *Brassicaceae* [32]. The higher number of *BrCMFs* compared to other species also reflects the WGT event in the *B.rapa*. Multiple *BrCMF* genes in *B. rapa* can jointly regulate the same physiological process.

The *BrCMF* genes differed greatly in sequence length as well as structure, and the intron–exon content varied considerably, indicating that members of the *CMF* family show high functional specificity and diversity. The phylogenetic trees of the *CMF* gene family in *B. rapa*, *A. thaliana*, and *O. sativa* have revealed that *CMF* genes can be divided into four groups based on their degree of developmental relatedness. The divergence of *OsCMFs* occurred earlier than that of *BrCMFs*, suggesting that the differentiation of *BrCMFs* occurred later than that of *OsCMF* in the monocotyledonous plant *O. sativa* [4]. Only two pairs of *OsCMF* and *BrCMF* genes showed high covariance, whereas more than 30 pairs of *AtCMF* and *BrCMF* genes showed high covariance. This 15-fold increase in the latter suggests that *B. rapa* is more closely related to *A. thaliana* and that they share homologous genes.

Plant hormones play a key role in regulating the stress resistance of plants [33]. ABA is a typical plant hormone associated with abiotic stress. It plays an important role in the responses to external stress signals to promote the expression of resistance-related genes [34,35,36] and regulate stomatal closure to limit water loss [37], which helps plants adapt to drought stress [37,38] and salt stress [37,39,40]. *MtCMFs* have been shown to regulate drought and salt stress either through ABA-dependent or ABA-independent mechanisms [10]. By regulating osmotic ions or transmitting signals via ABA [41], methyl jasmonate (MeJA) plays an important role [42] in regulating resistance to salt stress, drought stress [43,44], and low-temperature stress [26,45]. Gibberellin (GA) regulates the response to cold stress by inhibiting protein degradation via the activation of DELLA protein [46], and the accumulation of DELLA protein can reduce ROS levels, promote stress tolerance [47], and mediate ET signal transduction to promote salt tolerance [48]. The promoter regions of some *BrCMF* genes contain *cis*-acting elements, which is consistent with our findings; these genes are involved in abiotic stress resistance.

Both RNA-seq and RT-qPCR experiments revealed consistent changes in multiple genes, indicating that several genes within the *BrCMF* family are associated with different abiotic stress responses. *BrCMF22* is thought to respond to salt stress; *BrCMF14*, *BrCMF5*, and *BrCMF7* may regulate the response to drought stress; and *BrCMF21*, *BrCMF5*, and *BrCMF7* may respond to cold stress.

The expression of *BrCMF22* was high in flower and leaf tissues, and the promoter region is enriched with reactive oxygen species-inducing response elements associated with abiotic stress resistance and hormone-associated *cis*-acting elements. Additionally, there are anywhere from 1–8 response elements for ABA, jasmonic acid, and gibberellin, which are phytohormones that modulate salt stress signaling. The expression of some *CMF* genes was significantly downregulated under salt stress, such as *SiCCT31* in *S. italica* [11] and *AetCCT16* and *AetCCT20* in *A. tauschii* [14]. The expression of *BrCMF22* was also significantly downregulated under salt treatment according to RNA-seq and RT-qPCR analyses, as indicated by a study showing that the *Arabidopsis* homologous gene *AtCMF9* interacts with EMB1138 salt stress-related proteins. This indicates that *BrCMF22* regulates the salt stress response through hormone–protein interactions.

The results suggest that *BrCMF14*, *BrCMF5*, and *BrCMF7* may play a role in the regulation of drought stress. The transcriptome data indicated that *BrCMF14* was highly expressed in several organs, including roots, flowers, and fruits. Both transcriptome datasets indicated that drought stress reduced the expression level of this gene, and this was consistent with the expression patterns of *CMF* family genes in *A. tauschii* [14]. ABA and MeJA response elements in the promoter region of this gene were identified, and the Arabidopsis homolog *AtCMF10* interacts with PRR7, which regulates drought and ABA. Therefore, *BrCMF14* might play a regulatory role in drought stress by interacting with the PRR7 interaction protein and the ABA response element. In the roots and other organs, both homologous genes, *BrCMF5* and *BrCMF7*, were highly expressed. However, the expression of *BrCMF5* was significantly downregulated after drought treatment, whereas the expression of *BrCMF7* was significantly upregulated, which is consistent with the fact that both genes contain defense and stress-responsive elements, as well as MeJA-responsive elements. The expression of *GmCMF06* and *GmCMF07* [12] in *G. max* and *SiCCT3* [11] in *S. italica* was also significantly upregulated under drought stress. In addition to interacting with PRR7, the *Arabidopsis* homologous gene *AtCMF13* performs a similar function under drought conditions. *BrCMF7* contains an ABA response element, and PRR7 is regulated by ABA to regulate the response to drought stress [29], which might explain the upregulation of *PRR7* expression after drought treatment compared with *BrCMF5*.

The *B. rapa* genes *BrCMF21*, *BrCMF5*, and *BrCMF7*, which are homologous to *AtCMF13*, might be involved in the response to cold stress. RNA-seq and RT-qPCR analysis of these three genes revealed that cold treatment significantly reduced their expression levels, and the homologous gene *AtCMF13* interacted with the ambient temperature response protein APP6. Therefore, we hypothesized that *BrCMF21*, *BrCMF5*, and *BrCMF7* might be involved in the regulation of cold stress. There are two low-temperature-responsive *cis*-acting elements in *BrCMF5* and *BrCMF7*, as well as two−six MeJA-responsive elements in each of the three genes, and both *BrCMF21* and *BrCMF5* contain gibberellin elements, which impair cold tolerance by degrading the DELLA protein. Furthermore, these *cis*-acting elements suggest that these genes may also be involved in the response to cold stress, which is consistent with our findings.

## 4. Materials and Methods

### 4.1. Identification of CMF Genes in the B. rapa Genome

Based on previous research [4] and a search in the Phytozome database (Phytozome (doe.gov), 15 *AtCMF*s were obtained. The *B. rapa* genome data file was extracted from BRAD (http://brassicadb.cn/, accessed on 17 November 2022), and the *A. thaliana* genome data file was extracted from TAIR (http://www.Arabidopsis.org/, accessed on 16 November 2022). These genomes and TBtools (v1.120) [23] were then used to screen predicted genes based on their conserved domains, and genes without *CCT* domains were eliminated, with a total of 25 *CMF* genes being identified. Chromosome location information was retrieved from the genome annotation files in the *B. rapa* genome database. The length (aa), molecular weight (Da), and isoelectric point (pI) of the proteins encoded by the *CMF* genes were obtained from ExPASy [49] (https://web.expasy.org/protparam/, accessed on 21 November 2022). Subcellular localization data were obtained from PSORT (WoLF PSORT: Protein Subcellular Localization Prediction (hgc.jp)).

### 4.2. Chromosomal Localization, Synteny, and Phylogenetic Analysis

The locations of *CMF* genes on the *B. rapa* chromosome were determined from the *B. rapa* gff3 genome annotation information using TBtools (v1.120) software [23], while the chromosome density was calculated using the Gene Density Profile plug-in in TBtools (v1.120) with default settings. *BrCMF*s were named according to their chromosomal location.

Using the Dual Systeny Plot plug-in, Advanced Circos plug-in, and Table row extract or filter plug-in in the TBtools (v1.120) software, collinearity relationships between duplicate genes within *B. rapa* and between species were analyzed and visualized using data from *B. rapa*, *O. sativa*, and *A. thaliana*.

Phylogenetic trees of the complete amino acid sequences for *BrCMF*s, *AtCMF*s, and *OsCMF*s were constructed using the maximum likelihood method in MEGA X [50] with default parameters, while branch support was evaluated using 1000 bootstrap replicates. We used iTOL (https://itol.embl.de/ accessed on 27 November 2022) to make the phylogenetic tree.

### 4.3. Motif analysis, Gene Structure, and Conservative Domain Analysis

The Simple MEME Wrapper in TBtools (v1.120) was used to analyze conserved motifs using default parameter settings, with the exception of the Number of Motifs, which was set to 10. Conserved motif files were obtained from Batch CD-Search in NCBI (https://www.ncbi.nlm.nih.gov/, accessed on 22 November 2022) and visualized using the Visualize Pfam Domain Pattern tool (from Pfam Search) in TBtools.

### 4.4. Cis-Acting Element Analysis

The GXF Sequences Extract tool in TBtools (v1.120) was used to extract *BrCMF* promoter sequences 2000 bp upstream of the CDS, and the PlantCARE database [51] (PlantCARE, a database of plant promoters and their *cis*-acting regulatory elements (https://www.ugent.be/) accessed on 22 November 2022) was used for predictive analysis with default parameters. The results were visualized using TBtools (v1.120).

### 4.5. Tissue Specific Expression

Tissue-specific expression data for callus, flower, leaf, root, silique, and stem tissues from BRAD (http://brassicadb.cn/, accessed on 22 November 2022) were analyzed using the HeatMap plug-in of TBtools (v1.120).

### 4.6. Stress Treatments, Total RNA Extraction, Transcriptome and RT-qPCR Analysis

Seeds were sown in an MS-modified medium (containing vitamins, sucrose, and agar) (PM10121-307, Coolaber, Beijing, China) in a plant incubator. Six-leaved seedlings with similar growth statuses were obtained, and the seedlings were placed in a hydroponic system under salt stress (150 mM NaCl), drought stress (15% PEG6000), and cold stress (4 °C), while control (CK) seedlings were treated with dd H_2_O and grown at 25 °C/15 °C, with a light/dark cycle of 16/8 h. Materials that had been treated with salt stress for 12 h, drought stress for 6 h, and cold stress for 7 d were used for transcriptome analysis, *BrCMF* genes were subjected to Paired-end (PE) sequencing using Next-Generation Sequencing (NGS) based on the Illumina HiSeq sequencing platform by BioMarker Technologies (Beijing, China), and three biological replicates were collected for each sample. RT-qPCR experiments were conducted using samples that had been exposed to salt and drought stress treatment for 4 h, 6 h, and 12 h and exposed to cold stress treatment for 4 h, 6 h, 12 h, and 7 days. A minimum of three biological replicates were performed for each treatment, and samples were stored at −80 °C during storage.

Total RNA was extracted using a FastPure^®^ Cell/Tissue Total RNA Isolation Kit V2 (Vazyme Biotech Co., Ltd., Nanjing, China). Primer sequences were designed using the qPrimerDB-qPCR Primer Database (https://biodb.swu.edu.cn/qprimerdb/, accessed on 19–20 November 2022). RT-qPCR primer sequences are shown in Appendix A. TransScript^®^Uni All-in-One First-Strand cDNA Synthesis SuperMix was used for the reverse transcription of RNA samples for RT-qPCR analysis. RT-qPCR was performed using TransStart^®^Green qPCR SuperMix (TransGen Biotech, Beijing, China) on a qTOWER3 qPCR machine, with *BrACTIN2* as the internal reference gene, while the reaction was performed in three technical replicates. The data were analyzed using the 2^−∆∆CT^ method [52] and plotted using Excel 2020.

### 4.7. PPI Network Analysis

PPI network analysis was conducted using STRING (https://cn.string-db.org/, accessed on 23 November 2022) with default parameters, and Cytoscape v3.9.1 [53] was used to construct the interaction network.

## 5. Conclusions

A total of 25 *BrCMF* genes were identified in *B. rapa*, and some of these genes were involved in the regulation of abiotic stresses responses. *BrCMF22* might play a role in the response to salt stress, *BrCMF14*, *BrCMF5*, and *BrCMF7* might play a role in the response to drought stress, and *BrCMF21*, *BrCMF5*, and *BrCMF7* might play a role in the response to cold stress. Additional research is needed to clarify the specific mechanism underlying the observed responses.

## Figures and Tables

**Figure 1 plants-13-01118-f001:**
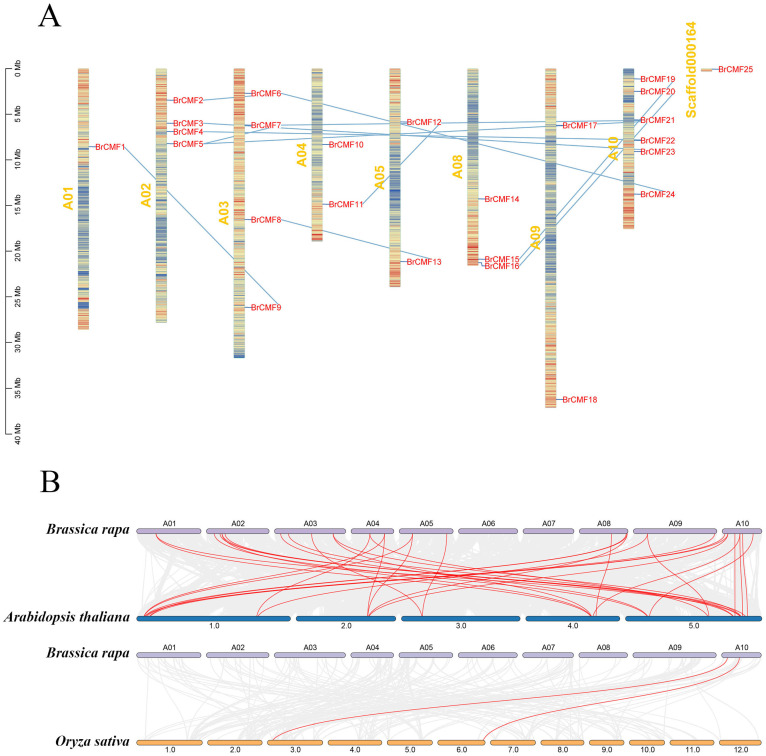
(**A**) The location of *BrCMF*s on *A. thaliana* chromosomes, with chromosomes indicated in yellow font, gene names in red font, and blue lines indicating collinearity relationships among genes. (**B**) Chromosomal collinearity relationships between *B. rapa* and *A. thaliana* and between *B. rapa* and *O. sativa*. Purple, blue, and orange indicate *B. rapa*, *A. thaliana*, and *O. sativa*, respectively, with red lines indicating collinearity relationships, while the gray lines represent orthologous gene blocks between Brassica rapa and other species. (**C**) The phylogenetic tree of *CMF* genes of three species: *B. rapa*, *A. thaliana*, and *O. sativa*. Yellow, green, blue, and purple correspond to Group 1, Group 2, Group 3, and Group 4, respectively. Other names of the genes are shown in parentheses.

**Figure 2 plants-13-01118-f002:**
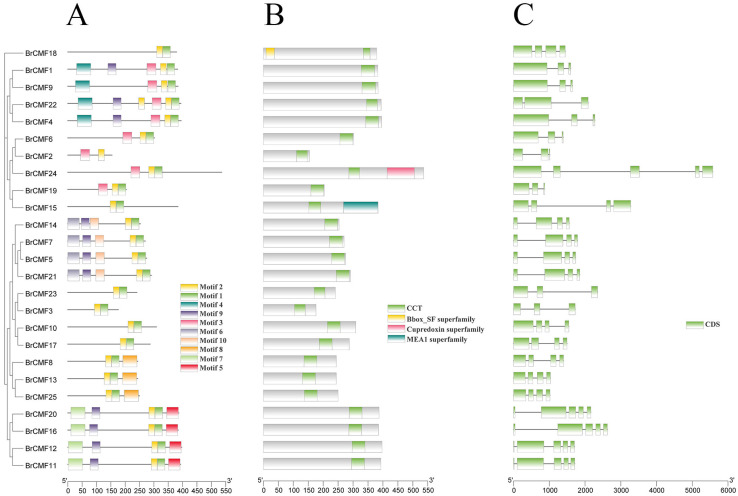
(**A**) The conserved motifs of *BrCMF*s, (**B**) the domains of *BrCMF*s, and (**C**) gene structure. The evolutionary relationships among the 25 genes are shown on the left.

**Figure 3 plants-13-01118-f003:**
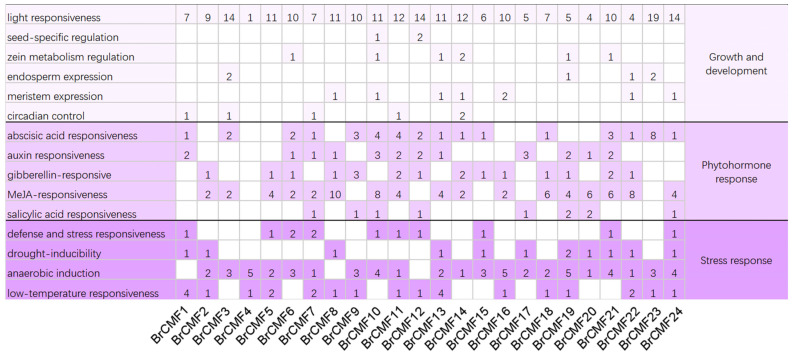
A total of 24 *BrCMF*s contain various *cis*-acting elements, and these *cis*-acting elements were classified into three groups based on their functions, including growth and development, phytohormone response, and stress response, as indicated on the right side of the figure. The functions of the *cis*-acting elements are shown on the left side, along with the number of *cis*-acting elements in each gene, while blank cells indicate the absence of the *cis*-acting element. Each column corresponds to a gene, which is indicated at the bottom, and each number indicates the number of cis-acting elements present in the 2000 bp 5’ region of the corresponding gene.

**Figure 4 plants-13-01118-f004:**
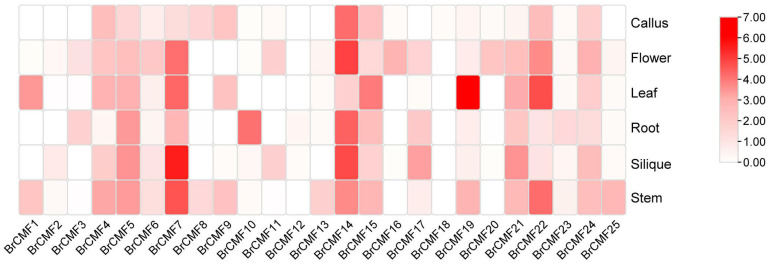
Heatmap of the tissue-specific expression patterns of *BrCMF* genes in various organs (Appendix A). Colors in the graph indicate the level of expression, light colors represent low expression levels, while red represents high expression levels. The data were all log-transformed to enhance contrast.

**Figure 5 plants-13-01118-f005:**
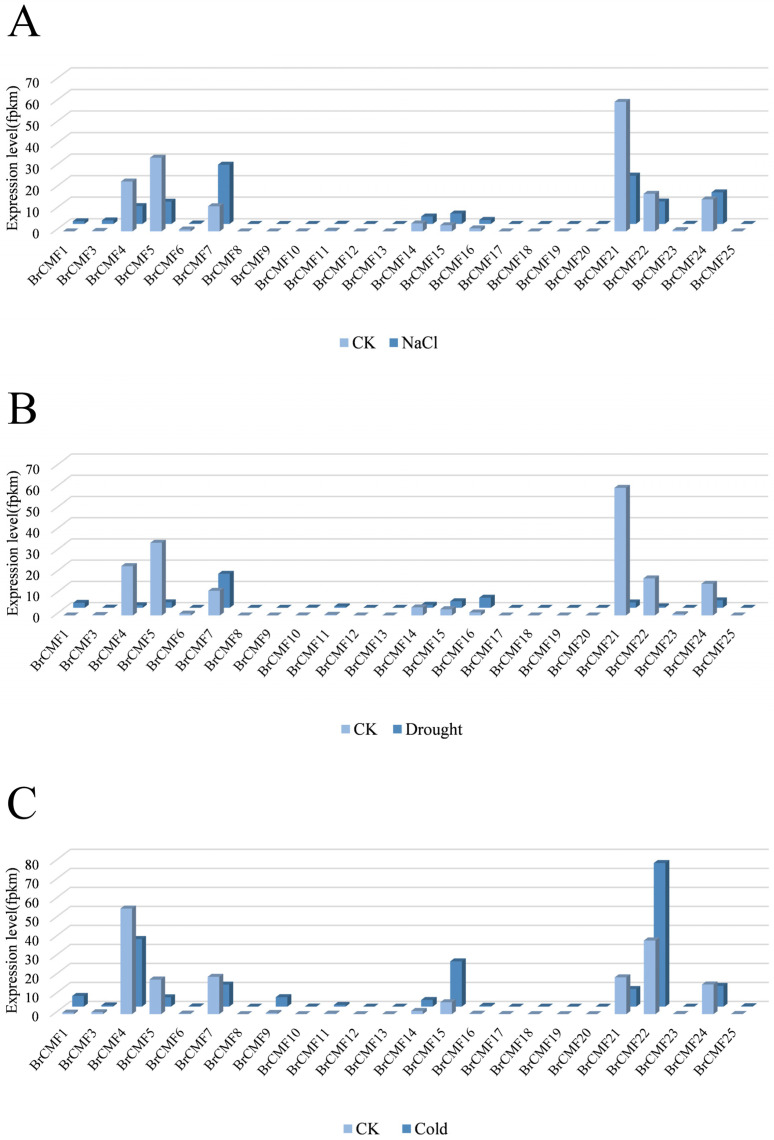
Transcriptome and RT-qPCR analysis reveals stress-induced changes in the expression of *BrCMFs* (*BrACTIN2* was used as reference for estimated ∆CT in Figure 5A–C). (**A**) The expression levels of *BrCMFs* transcriptome after 12 h of NaCl treatment. (**B**) The expression levels of *BrCMFs* transcriptome after 6 h of PEG treatment. (**C**) The expression of *BrCMFs* transcriptome after 7 d of cold treatment. The height of column represents the expression level. (**D**) The expression levels of *BrCMFs* according to RT-qPCR analyses after 0 h (CK), 4 h, 6 h, and 12 h of NaCl and PEG treatments. (**E**) The expression levels of *BrCMFs* according to RT-qPCR analyses after 0 h (CK), 4 h, 6 h, 12 h and 7 d of cold treatments. The expression of *BrCMF5* and *BrCMF21* decreased to 0 after 7 days of cold stress. Error bars are indicated for each column. CK refers to 0 h treatment, 7 d refers to 7 days. PEG refers to the PEG6000 treatment. Full details are given in Appendix A.

**Figure 6 plants-13-01118-f006:**
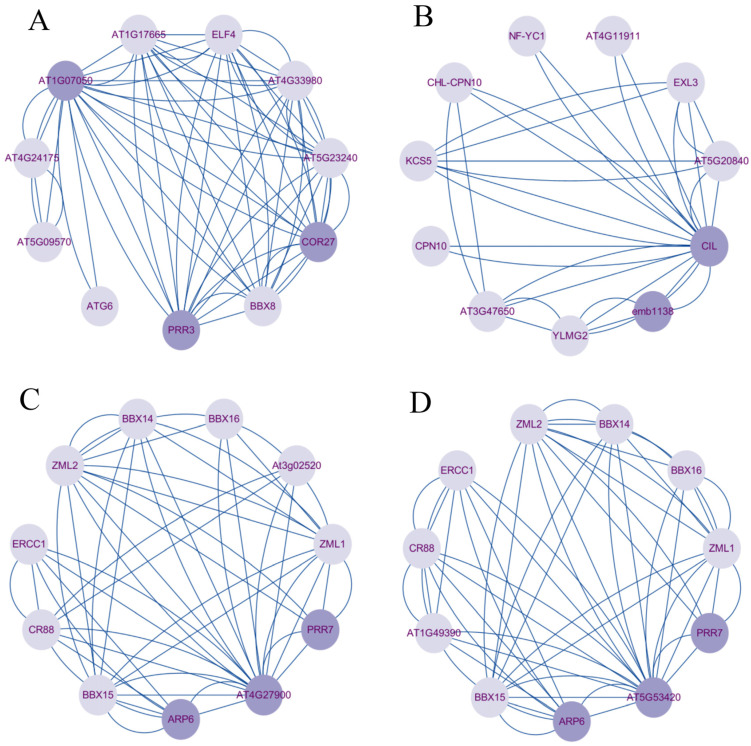
The PPI networks of four *A. thaliana* homologous genes: (**A**) *AtCMF3* (*AT1G07050*), (**B**) *AtCMF9* (*CIL*), (**C**) *AtCMF10* (*AT4G27900*), and (**D**) *AtCMF13* (*AT5G53420*). At the network nodes, the circles represent proteins, dark purple circles represent homologous genes and proteins associated with abiotic stress, light purple circles represent other interacting proteins, and connecting lines indicate associations between proteins.

**Table 1 plants-13-01118-t001:** Physicochemical properties of *BrCMF*.

Gene Name	Gene ID	Chromosome	pI	MW (Da)	Protein (aa)	Subcellular	*A. thaliana* ID	*A. thaliana* Name
*BrCMF1*	*Bra013939*	A01:8554984-8556579	6.37	42,602.83	382	Nuclear	*AT4G25990*	*AtCMF9*(*CIL*)
*BrCMF2*	*Bra023461*	A02:3453187-3454201	7.79	17,412.59	153	Nuclear, Cytosol	*AT5G14370*	*AtCMF11*
*BrCMF3*	*Bra020271*	A02:5974545-5976267	8.81	19,452.58	175	Nuclear	*AT5G59990*	*AtCMF15*
*BrCMF4*	*Bra020461*	A02:6894679-6896952	7.73	43,853.06	394	Nuclear	*AT5G57180*	*AtCMF14*(*CIA2*)
*BrCMF5*	*Bra022655*	A02:8231286-8233020	5.16	30,843.57	274	Nuclear	*AT5G53420*	*AtCMF13*
*BrCMF6*	*Bra006252*	A03:2698486-2699873	9.34	33,962.85	301	Nuclear	*AT5G14370*	*AtCMF11*
*BrCMF7*	*Bra029069*	A03:6182682-6184469	4.77	30,419.14	269	Nuclear	*AT5G53420*	*AtCMF13*
*BrCMF8*	*Bra001475*	A03:16536050-16537448	5.9	28,039.74	243	Nuclear	*AT3G12890*	*AtCMF8 (ASML2)*
*BrCMF9*	*Bra019134*	A03:26145861-26147506	6.22	42,959.09	383	Nuclear, Chloroplast, Mitochondrion	*AT4G25990*	*AtCMF9*(*CIL*)
*BrCMF10*	*Bra025502*	A04:8024942-8029130	5.33	34,785.22	308	Nuclear	*AT5G41380*	*AtCMF12*
*BrCMF11*	*Bra021846*	A04:14851491-14853202	4.45	43,463.76	392	Nuclear	*AT2G33350*	*AtCMF6*
*BrCMF12*	*Bra005503*	A05:5839671-5841373	4.68	44,285.42	396	Nuclear	*AT2G33350*	*AtCMF6*
*BrCMF13*	*Bra034721*	A05:21150216-21151247	5.09	27,909.4	243	Nuclear	*AT3G12890*	*AtCMF8 (ASML2)*
*BrCMF14*	*Bra010391*	A08:14276724-14278279	5.06	28,599.34	252	Nuclear	*AT4G27900*	*AtCMF10*
*BrCMF15*	*Bra030669*	A08:20873312-20876588	4.56	43,689.41	383	Nuclear, Cytosol	*AT1G07050*	*AtCMF3*(*FITNESS*)
*BrCMF16*	*Bra030574*	A08:21264649-21267272	4.61	42,706.8	384	Nuclear	*AT1G04500*	*AtCMF1*
*BrCMF17*	*Bra027785*	A09:6208655-6210149	5.05	32,671.79	286	Nuclear	*AT1G63820*	*AtCMF4*
*BrCMF18*	*Bra032471*	A09:36209798-36211242	5.83	43,399.58	378	Nuclear	*AT1G05290*	*AtCMF2*
*BrCMF19*	*Bra015548*	A10:1127207-1128077	5.06	24,336.74	203	Nuclear	*AT1G07050*	*AtCMF3*(*FITNESS*)
*BrCMF20*	*Bra015321*	A10:2486571-2488734	4.48	43,148.13	386	Nuclear	*AT1G04500*	*AtCMF1*
*BrCMF21*	*Bra003057*	A10:5619206-5621057	4.68	32,644.78	290	Nuclear, Cytosol, Mitochondrion	*AT5G53420*	*AtCMF13*
*BrCMF22*	*Bra002752*	A10:7861017-7863103	8.95	43,930.57	393	Nuclear	*AT4G25990*	*AtCMF9*(*CIL*)
*BrCMF23*	*Bra002516*	A10:9083709-9086058	6.3	26,425.3	240	Nuclear	*AT5G59990*	*AtCMF15*
*BrCMF24*	*Bra008763*	A10:13747298-13752870	9.67	60,150.03	535	Nuclear, Chloroplast	*AT5G14370*	*AtCMF11*
*BrCMF25*	*Bra039351*	Scaffold000164	5.61	28,406.09	249	Nuclear	*AT3G12890*	*AtCMF8 (ASML2)*

Basic information of *BrCMF*s is shown in the above table, and the italics in parentheses in the *A. thaliana* column are the alternate names of the genes. “MW” is molecular weight and “pI” is isoelectric point.

## Data Availability

Data are contained within the article/Appendix A.

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
