# Peer review of "Genome-Wide Identification of BrCMF Genes in Brassica rapa and Their Expression Analysis under Abiotic Stresses"

_plants, 2024, doi:10.3390/plants13081118_

Round 1

Reviewer 1 Report

Comments and Suggestions for Authors

Dear authors,

I had the pleasure to read the manuscript entitled "Genome-wide identification of BrCMF genes in Brassica rapa and their expression analysis under abiotic stress".

I found it quite interesting but with some lacks in the results section.

Fig.1B: what are the grey lines?
It is not fully clear the sense of the sentences on the lines 196-197 Most genes in this group had two or three cis-acting elements, suggesting that some of these genes play an important role in stress responses: I could see that many genes has elements in the section of Phytohormone response. and many others in the "light responsiveness": How do the authors explain that?
In the paragraph of Tissue specificity, it is not clear the specificity: In most tissues, the expression of the genes 210
was low; however, the expression of nine genes was high in some tissues and low in oth- 211
ers. Genes with similar sequences tended to have similar expression levels. Are the expression levels low or high? Not clear however it is clear that they play a role in several life activities.

The authors are invited to complete  the captions because they are not well described.

Line 211: the expression of nine genes was high in some tissues.... where one can see these data?

Fig.4: where the data come from?

The main question is: why the transcriptomic data come from 12h of treatments and the qPCR from different treatment times? If the authors don't get the same results from the two experiments it is because they have got two different experiments. Due to the importance of the results and of the discussion, could the authors provide the results from two similar experiments?

In the 3 plots in the FIg.5 (A, B, C) no statistics is reported but needed to my opinion due to the prticular plots showed.

line 235-236: it is not clear which genes the authors are talking about.

line 253-254: it not clear what they autors are talking about, maybe they could be more precise?

In general, to my opinion the manuscript is not yet ready for publishing. It is full of nice and interesting results but needs to be accurately improved.

Thank you

Reviewer 2 Report

Comments and Suggestions for Authors

The report contains interesting information for the plant biology community and has good potential to eventually be published. However, several aspects must be addressed promptly before considering publication again. In addition to minor editorial errors, my main concerns about the manuscript are indicated in the attached document. Something that caught my attention is that according to Figure 1A, the species under study has 8 chromosomes, but in Figure B, the collinearity relationships were made with 10 chromosomes.

Other aspects that need to be clarified are the obtaining and processing the transcriptomic data, what equipment and strategy was used for sequencing, what was the number of reads and how many genes were identified as differentially expressed. Particularly, I believe that a more detailed description and analysis of the transcriptomic data would substantially raise the quality of the report. In its current version, it is also necessary to clarify why the conditions to analyze expression by RT-qPCR were not the same as those used to create the transcriptomes.

Comments on the Quality of English Language

English is not my native language and beyond the editorial errors pointed out in the attached document, I do not feel qualified to make a more detailed analysis of it. Although for me the manuscript is understandable.

Reviewer 3 Report

Comments and Suggestions for Authors

This manuscript analyses the whole gene subfamily CMF in the cruciferous plant Brassica rapa. Most of result are analysis in silico in a wide variety of bioinformatics plants tools. Experimentally, the manuscript is poor because only present a transcriptomic and expression analysis. The paper is well writing and the discussion was well done. The topic is the general interest for science community but the objective must be more ambitious. In my modest opinion the paper is well planning but the experimental results are limited. Some interactomic experiments could be made, overall with the genes that showed more differences in gene expression in response to abiotic stress. The hypothetical interaction proposed in silico could be demonstrated with interaction assays (immunoprecipitation, yeast double hybrid…etc). After read the paper I have the feeling that the manuscript are unfinished. For this reason, I consider that it will require a mayor revision of your manuscript for publishing in Plant. I suggest the following considerations and comments:

Introduction

L66-71.- This sentence is confused and must be review. You must indicate the number of genes in A. thaliana and explain why these four genes are especial.

L79.-   “affect B. rapa yield” could be change by “affect negatively B. rapa yields”.

Results

L101.- The protein length has units, amioacids (AAs).

L102.- your repeat predicte, it could be change by hypothetical genes.  

L102-105.- In this sentence, I don’t know if you make reference to gen or protein location. First you said “genes predicted to be located” …. But in the citosol there no genes, maybe you wanted to maka reference to gene product.

L165 .- There are two full stops.  

In some sentence you abuse of semicolon in sentences with any semantic relation. Please review this aspect of the writing.

The point results 2.6 about transcriptomic and expression analysis is not well structured and is difficult to understand it. You must indicate the figure and chart letter in the text to make easy the reading. While I was reading this paragraph I was not clear if the results belong to the transcriptomic analysis or RT-qPCR assays. Please review and rewrite this point.

In the figure 5, the charts A-C must indicate what sample was used as reference for estimated deltadeltaCt. In the chart D is clear that the control sample (CK) was used as reference for every gen. In the expression charts the Y axis label must to be Fold Change. CK means must to be describe in the caption. In the chart 5D, NACL must to be change by NaCl (this is chemical nomenclature), PEG is a chemical polymer used to induce osmotic or drought stress so I would change PEF in the X axis label by drough.

L262.- under must to be change by after.

Discussion

I would like that you discuss why in B. rapa ther are the highest number of CMF genes.

L309-311.- This sentence is incongruent for me, please review. You said “in the dicotyledonous plant O. Sativa”, the rice is a cereal belonging to the monocotyledonous plant group.

L354.- In both the roots must to be change by In the root.

Materials and Methods.

The experimental planning of the transcriptomic analysis is questionable. For example: You induce salt stress treated the plant for 12 hours. After 12 hours is difficult to find any symptom of ionic stress due to ions accumulation,  only  a osmotic stress could be detected at short time. For this reason, the results are similar under salt and drought stress. Please read Annual Review abut salt stress of Rana Munn and Mark Tester.  

Round 2

Reviewer 1 Report

Comments and Suggestions for Authors

Dear authors,

thank you for making the manuscritp clearer. I will recommend this version the the editors to be published.

Kind regards

Author Response

We thank the reviewers for recognizing our manuscript! We have changed "abiotic stress" to "abiotic stresses" at the suggestion of the Academic Editor to make the manuscript more rigorous!